# The Effect of Citral on Aquaporin 5 and Trpv4 Expressions and Uterine Contraction in Rat—An Alternative Mechanism

**DOI:** 10.3390/life11090897

**Published:** 2021-08-30

**Authors:** Adrienn Seres-Bokor, Kata Kira Kemény, Hoda Taherigorji, Annamária Schaffer, Anna Kothencz, Róbert Gáspár, Eszter Ducza

**Affiliations:** 1Department of Pharmacodynamics and Biopharmacy, Faculty of Pharmacy, University of Szeged, Eötvös u. 6., 6720 Szeged, Hungary; seres-bokor.adrienn@szte.hu (A.S.-B.); kemeny.katakira@gmail.com (K.K.K.); hoda.taherigorji@gmail.com (H.T.); 2Department of Pharmacology and Pharmacotherapy, Albert Szent-Györgyi Medical School, University of Szeged, Dóm tér 12., 6721 Szeged, Hungary; schaffer.annamaria@med.u-szeged.hu (A.S.); kothencz.anna@med.u-szeged.hu (A.K.); gaspar.robert@med.u-szeged.hu (R.G.)

**Keywords:** aquaporin 5, TRPV4, citral, pregnancy, preterm birth

## Abstract

Aquaporins (AQPs) are expressed in the uterus, playing a physiological role during pregnancy. An osmotic pathway—through AQP5—may modify the transient potential vanilloid 4 (TRPV4) function and uterine contraction. Our aim was to determine the role of TRPV4 antagonist citral in the regulation of pregnant uterine contraction. In vitro uterine contractions were evoked by KCl and the response was modified with citral. The expressions of TRPV4 and AQP5 were measured by RT-PCR and Western blot techniques. The lengths of gestational periods were determined in normal and LPS-induced preterm births after citral treatment, in vivo. Citral significantly decreased the uterine contraction on day 22 of pregnancy. AQP5 expression significantly increased after citral incubation; however, TRPV4 expression did not show significant changes. After citral pretreatment, the gestational period was extended both in normal and LPS-induced preterm births. Our results suppose that the downregulation of AQP5 may initiate hypertonic stress, activating TRPV4 the uterine contraction on the last day of the gestational period. The putative cooperation between AQP5 and TRPV4 may open a novel target to treat or prevent preterm birth.

## 1. Introduction

Aquaporins (AQPs) are crucial for water and small solute transport across physiological barriers. Several AQP families have already been identified (AQP0 to AQP12) [1]. AQPs play role during gestational period including parturition, myometrial contractility, and cervical ripening [2]. In our previous studies, AQP5 expression was found to be predominant during pregnancy in rats, although it was significantly downregulated at the last gestational day. Moreover, AQP5 was upregulated by oxytocin [3] and its expression was estrogen and progesterone dependent in the uteri of late-term rats [4]. It is supposed that at the end of pregnancy the intensity of the uterine contractions has an inverse correlation with the expression of the uterine AQP5.

The uterine Ca^2+^ level is essential for contractions, the myometrial Ca^2+^ influx is the main drive for spontaneous rhythmic contractions [5]. Transient receptor potential (TRP) ion channels are nonselective cation membrane channels. The TRP subfamily transient receptor potential vanilloid (TRPV) channels are permeable to calcium, magnesium, and sodium [6]. TRPV4 is the fourth cloned member of the TRPV subfamily reported as a modulator of uterine contractility in rats [7,8]. TRPV4 can be activated by diverse stimuli, including osmotic or mechanical stimuli, warm temperature, phorbol ester derivatives, products of arachidonic acid breakdown or feedback regulation by Ca^2+^ [9]. As the prominent co-localization of TRPV4 and AQP4 in retinal Müller glia cells was earlier identified by Ryskamp DA et al. [10], and we also determined the co-expression of TRPV4 and AQP5 in late-pregnant rat uterine tissue [7]. 

Citral (3,7-dimethyl-2,6-octadienal) is the major aromatic component of lemongrass oil. It is commonly used in daily life; for instance, as a flavor to intensify taste, or as a fragrance or as a repellent. It exhibits various pharmacological effects, such as a hepatoprotective effect in nonalcoholic fatty liver disease or an antimicrobial effect, but it also has immunomodulatory and antihyperalgesic effects [8,11,12,13]. Citral produces a relaxant effect on K^+^ and phenylephrine-induced contractions of isolated aorta rings in rats [14]. Furthermore, it is a well-known antagonist for TRPV4 [15]. 

The aim of our study was to investigate the effect of citral on the cooperation of AQP5 and TRPV4 in the late-pregnant rat uterus in vitro and in vivo to identify their mutual influences on myometrial contraction (Figure 1).

## 2. Materials and Methods

### 2.1. Housing and Handling of the Animals

The animals were treated in accordance with the European Communities Council Directive (2010/63/EU) and the Hungarian Act for the Protection of Animals in Research (Article 32 of Act XXVIII). All experiments involving animal subjects were carried out with the approval of the National Scientific Ethical Committee on Animal Experimentation (registration number: IV/2767/2020). Sprague–Dawley rats (Animalab Ltd., Vác, Hungary) were kept at 22 ± 3 °C; the relative humidity was 30–70% and the light/dark cycle was 12/12 h. The animals were maintained on a standard rodent pellet diet (INNOVO Ltd., Gödöllő, Hungary) with tap water available *ad libitum*. The rats were terminated by CO_2_ inhalation.

### 2.2. Mating of the Animals

Mature (10 weeks old), female (180–200 g) and male (240–260 g) Sprague-Dawley rats were mated in a mating cage before dawn. Within 4 h after the potential mating, vaginal smears were taken and investigated under the microscope (magnification rate 1000×). If sperms or vaginal plugs were found, the female rats were separated as first-day pregnant animals.

### 2.3. Experimental Mode

Figure 1 summarizes our experimental model. The rats got in vivo citral treatments and then their uteri were involved into in vitro studies. Additionally, a group of treated rats were allowed to deliver their pups to measure the effect of citral treatment on the length of gestational period. 

### 2.4. In Vitro—Isolated Contractility Study

The uteri of 22-day pregnant Sprague-Dawley rats were dissected, the fetuses were removed, and the uteri were sliced into 0.5-cm-long rings. The smooth muscle tissues were mounted vertically in an organ bath containing 10 mL of de Jong solution (composition in mM: 137 NaCl, 3 KCl, 1 CaCl_2_, 1 MgCl_2_, 12 NaHCO_3_, 4 NaH_2_PO_4_, 6 glucose, pH 7.4). The temperature of the organ bath was maintained at 37 °C with a continuous carbogen (95% O_2_ + 5% CO_2_) supply. 

The initial tension of the uterine rings was set about 1.5 g, the equilibration time was 60 min with a buffer change every 15 min. The contractions of the 22-day myometrial rings were measured with a gauge transducer (SG-02) and recorded with a SPEL Advanced ISOSYS Data Acquisition System (MDE Ltd., Budapest, Hungary). The rhythmic uterine contractions were elicited by 25 mM KCl and the cumulative concentration-response curve of citral (10^−8^–10^−2^ M) was obtained. 

### 2.5. Studies with Citral Pretreatment

The 22-day pregnant uterus samples were incubated in the isolated organ bath chambers with 10 μM citral for 2 h. After citral incubation, the uteri were removed for RT-PCR and Western blot measurements.

### 2.6. In Vivo Studies—Investigation of the Length of the Gestational Period

The animals were divided into 6 groups (*n* = 6–8): (**1**) intact pregnant rats on day 22 of pregnancy (normal term labor, *n* = 16); (**2**) pregnant rats, orally treated with olive oil, as a vehicle control, on days 18–22 once daily; (3) pregnant rats, orally treated with citral (240 mg/kg [3], administered by oral gavage; in olive oil vehicle; Sigma-Aldrich, Budapest, Hungary) on days 18–22 once daily; (**4**) pregnant rats treated with LPS (50 μg/animal, intraperitoneally; Sigma Aldrich, Hungary) on day 18 of pregnancy to induce preterm birth on day 20 [16]; and (**5**) pregnant rats treated with LPS on day 18 and citral on days 18–22 once daily. The administration protocol is outlined in Table 1. The rats were closely observed for any signs of preterm delivery (e.g., decreased movement, nest-building behavior or vaginal bleeding). The beginning of labor was defined by the delivery of the first pup. 

### 2.7. RT-PCR Studies

*Tissue isolation*. Pregnant uteri samples were placed in RNAlater Solution (Sigma-Aldrich, Hungary), then stored at −75 °C until the extraction of total RNA.

*Total RNA preparation from tissue*. Total cellular RNA was isolated by extraction with guanidinium thiocyanate-acid-phenol-chloroform by the protocol of Chomczynski and Sacchi [17]. After precipitation with isopropanol, the RNA was washed with 75% ethanol and resuspended in diethyl pyrocarbonate-treated water. RNA purity was measured at an optical density of 260/280 nm with BioSpec Nano (Shimadzu, Kyoto, Japan); all samples exhibited an absorbance ratio in the range of 1.6–2.0. RNA quality and integrity were assessed by agarose gel electrophoresis. 

*Real-time quantitative reverse transcription-PCR (RT-PCR).* Reverse transcription and amplification of the PCR products were performed by using the TaqMan RNA-to-C_T_-Step One Kit (Thermo Fisher Scientific, Budapest, Hungary) and an ABI StepOne Real-Time cycler. Reverse-transcriptase PCR amplifications were performed as follows: at 48 °C for 15 min and at 95 °C for 10 min, followed by 40 cycles at 95 °C for 15 s and at 60 °C for 1 min. The samples of qPCR experiments contained “no-template” control or RNA samples from non-treated and treated uterus. The generation of specific PCR products was confirmed by melting curve analysis. The following primers were used: assay ID Rn00562837_m1 for the *Aqp5* water channel, Rn00576745_m1 for *Trpv4* and Rn00667869_m1 for *β-actin* as endogenous control (Thermo Fisher Scientific, Hungary). All samples were run in triplicate. The fluorescence intensities of the probes were plotted against PCR cycle number. The amplification cycle displaying the first significant increase in the fluorescence signal was defined as the threshold cycle (C_T_).

### 2.8. Western Blot Analysis

The uterine tissues from pregnant animals (tissue between two implantation sites) were homogenized using a Micro-Dismembrator (Sartorius AG, Göttingen, Germany) and centrifuged at 5000× *g* for 15 min at 4 °C in RIPA Lysis and Extraction Buffer (Thermo Fisher Scientific, Hungary) with protease and phosphatase inhibitor cocktail. Total protein amounts from supernatant were determined with spectrophotometry (BioSpec-nano, Shimadzu, Japan).

25 μg of protein per well was used for electrophoresis on 4–12% NuPAGE Bis-Tris Gel in XCell SureLock Mini-Cell Units (Thermo Fisher Scientific). Proteins were transferred from gels to nitrocellulose membranes by iBlot Gel Transfer System (Thermo Fisher Scientific). Antibody binding was detected with the WesternBreeze Chromogenic Western blot immunodetection kit (Thermo Fisher Scientific, Hungary). The blots were incubated overnight, at 4 °C, on a shaker with AQP5 (cat. no AB-15858, 1:200, Sigma-Aldrich), β-actin (cat. no bs-0061R, 1:200, Bioss Antibody) and TRPV4 (cat. no OSR00136W 1:200, Thermo Fisher Scientific) polyclonal primary antibody in the blocking buffer. The incubation of the secondary antibody solution was carried out based on the protocol of WesternBreeze^®^ Chromogenic Immunodetection Kit. Images were taken with the EDAS290 imaging system (Kodak Ltd., Budapest, Hungary), the optical densities of immunoreactive bands were determined with Kodak 1D Images analysis software. Optical densities were expressed as arbitrary units after the subtraction of the local area background.

### 2.9. Statistical Analysis

Statistical analyses were performed with the Prism 4.0 software (Graphpad Software Inc. San Diego, CA, USA). All data were analyzed by one-way ANOVA, followed by Dunnett multiple comparison test or Student’s *t*-test, and each is presented as the mean ± SEM. Significance was accepted at *p* < 0.05.

## 3. Results

### 3.1. Contractility Studies

Citral (10^−8^–10^−2^ M) inhibited the KCl-evoked contractile response in the 22-day pregnant rat uteri (Figure 2). In low (10^−8^–10^−5^ M) concentrations its relaxant effect was almost nonsignificant; however, in high concentrations (10^−4^–10^−2^ M) its action was increased steeply. The maximum relaxant effect of citral was 62.9% ± 2.5, while the EC_50_ was 2.16 × 10^−4^ ± 6.07 × 10^−5^ M.

### 3.2. Studies with Citral Pretreatment

AQP5 and TRPV4 mRNA and protein expression were measured in the 22-day pregnant uteri after citral treatment (10 μM for 2 h), in vitro. AQP5 expression significantly increased after citral incubation compared to the control (Figure 3A,B). However, the changes in TRPV4 expression were not significant (Figure 3C,D).

### 3.3. Effect of Citral Treatment on the Length of the Gestational Period

The time of birth of the control rats mostly occurred during the morning (t = 11.9 ± 0.5 h) on the 22nd day of pregnancy. The length of the gestational period significantly increased (mean time of birth on day 22: 15.9 ± 0.9 h) after vehicle (olive oil) treatment. The gestational period was further extended after citral treatment (t = 28.6 ± 2.3 h) in normal pregnancy (Figure 4A). During the experiments Side effects or toxic effects were not observed in any of the groups.

The time of labor in inflammatory-induced preterm birth (LPS) was during the night (t = 0.8 ± 1.5 h) on day 20. No significant changes were determined in the time of labor after vehicle treatment in LPS rats (t = 1.6 ± 1.0 h). The gestational period of LPS-treated rats was significantly longer, the time of delivery was extended to the dawn of the 22nd day of pregnancy (t = 54 ± 5.5 h) after citral treatment (Figure 4B).

## 4. Discussion

The annual number of premature babies is estimated 15 million [18] and preterm birth (PTB) is responsible for most of the infant mortality and morbidity. Since the causes of PTB are widely varying, therefore the identification and prevention of PTB is very difficult, sometimes impossible and their low chance to reduce the PTB rate in the near future. The solutions may arise by interdisciplinary efforts, gaining a better understanding of the pathomechanism of PTB [19].

In our previous study, an inverse correlation was confirmed between the AQP5 level and myometrial contraction, a reduced uterine AQP5 expression was accompanied with enhanced contractility. Several signal transduction mechanisms were identified to regulate AQP expressions in animals. An osmotic pathway, in which the changes in AQP5 expression are presumed to influence the osmotic milieu in late-pregnant uterine tissue, may lead to the alteration of the osmotic-sensitive TRPV4 channel modifying the myometrial Ca^2+^ level. This pathway was confirmed by the irreversible correlation between the expression of TRPV4 and AQP5 in the late-pregnant rat uterus [7]. 

Currently, in vivo experimental tests for AQPs are very limited because subtype-selective agents are not available. Hence we examined this system from the TRPV4 -“side” using a well-tolerated, natural plant product, citral.

Citral is an antioxidant, inhibitor of nuclear factor kappa B (NF-κB) and cyclooxygenase-2 (COX-2) and activator of peroxisome proliferator-activated receptor (PPAR)-α and γ. Beside these actions, it elicits a sustained inhibition of TRP channels of sensory neurons (like TRPV1–3 and TRPM8), and a short-term blockade of TRPV4 and TRPA1 [20]. In addition, it is an odorous ingredient in beverages, foods, and cosmetics. A novel study highlighted that citral has potential therapeutic significance as a smooth muscle relaxant and local anesthetic, as it promotes relaxation in tracheal and aortic smooth muscles and inhibits nerve excitability in animal models [21]. Since our recent work involved an in vivo experiment with pregnant rats. Therefore, we had to choose a TRPV4 inhibitor that is safe for pregnant animals, and its dosage and pharmacokinetic parameters are available. Citral met with these requirements [22]. Unfortunately, none of the available selective TRPV4 blockers e.g., GSK 2193874, RN-1734 has pharmacokinetic, gestational safety or in vivo dosage data. Therefore, we could not involve them in our in vivo experiment.

In our recent study, citral treatment-induced dose-dependent uterus relaxation in vitro and enhanced the length of normal pregnancy. Surprisingly, olive oil significantly extended the length of normal gestational period, but such an effect of citral was even higher. The uterus relaxant effect of the active compound of olive oil (oleocanthal) has been proved by the inhibition of calcium/PGF2-induced uterine hypercontraction [23]. Oleic acid supplementation favors basal PGE2 production in both the maternal endometrium and the fetal allantochorion. In the maternal endometrium, oleic acid increased the ratio of PGE2:PGF2α and antagonized the oxytocin-induced PG production. It is very probable that elevated oleic acid level may have an impact on parturition including the initiation and the progression as well [24]. This effect of olive oil could not be detected in LPS-induced preterm birth. Endotoxin treatment in rats induces systemic inflammatory responses both in the mother and the fetus [25]. In parallel, downregulation of AQP5 in LPS- and hormonally induced preterm birth were detected, too [4]. 

The anti-inflammatory and tocolytic effects of citral have already been investigated in vitro, it increased the cAMP level in human myometria [26]. An alternative pathway was investigated in our studies via the cooperation of AQP5 and TRPV4. External Ca^2+^ entry via voltage-gated calcium channels is essential for myometrial contractions. However, other calcium channels are also important in the contractile response. One potential candidate is the TRPV4 channel [27]. TRPV4 channel activity modulates uterine contractility in murine models [28] and human myometrial tissues, in vitro [29]. The functional interaction between the selective water channel AQP5 and ion channels, such as TRPV4, closely participates in the regulation of osmotic homeostasis. In salvia gland cells, TRPV4 has a proved interaction with AQP5. In lack of AQP5, the hypotonicity generated calcium flow via TRPV4 was significantly reduced and subsequently the decrease in regulatory volume ceased [30,31]. TRPV4 plays a role in the hypotonic reduction of AQP5, including the control of extracellular calcium. This is proved by the increase in intracellular calcium after hypotonic stimulation, and a decrease in AQP5 in the presence of TRPV4 stimulator [32]. 

Based on our earlier result, we assume that high AQP5 expression from pregnancy day 18 [3] resulted in a hypotonic cytoplasm in uterus cells, which triggers smooth muscle cell membrane stretching and activates calcium influx through TRPV4. This mechanism is confirmed by the colocalization of AQP5 and TRPV4 in late-pregnant rat uterus and the increasing expression of TRPV4 at the time of labor [7]. As a result of cell volume regulation, AQP5 expression decreased on the last day of pregnancy. 

The expression and activation of TRPV4 are modified along with volume expansion. Citral (TRPV4 antagonist) treatments suppressed cell volume expansion, thereby influenced AQP5 expression, inhibited uterus contraction and extended the duration of pregnancy in rat.

Our results give basic information about the role of AQP5 and TRPV4 in delivery and may provide new insight into the unrevealed process of preterm delivery. The clinical translation of our results might open a new possibility for the therapy and prevention of preterm birth targeting AQP5 and TRPV systems. Further preclinical studies are required with selective TRPV4 inhibitors; however, those studies must be preceded by pharmacokinetic and gestational safety measurements of these selective blockers in pregnant rats

## Figures and Tables

**Figure 1 life-11-00897-f001:**
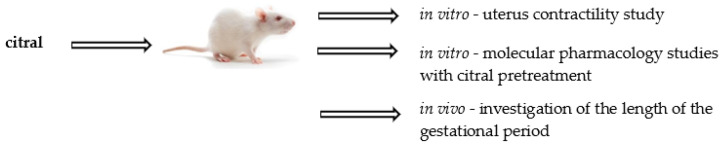
The summary of the experimental model for the citral-treated rats. After the treatments, animals were involved both into in vitro and in vivo studies.

**Figure 2 life-11-00897-f002:**
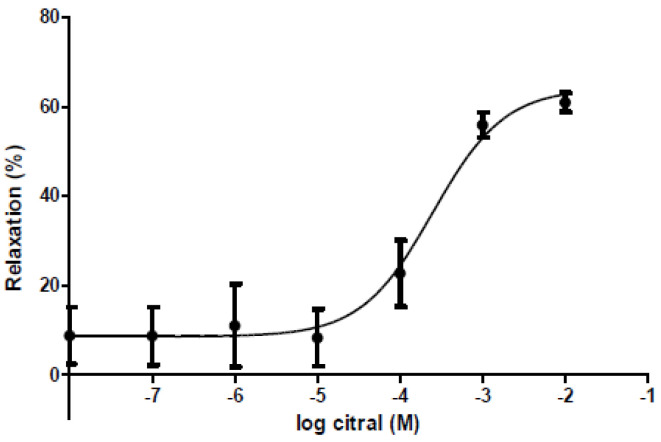
Cumulative concentration-response curve of citral (10^−8^–10^−2^ M) on 22-day pregnant rat uteri. The alteration in contraction was determined by the area under the curve (AUC) method and expressed in % ± S.E.M. as compared with the AUC of KCl-evoked control contractions. *n* = 6.

**Figure 3 life-11-00897-f003:**
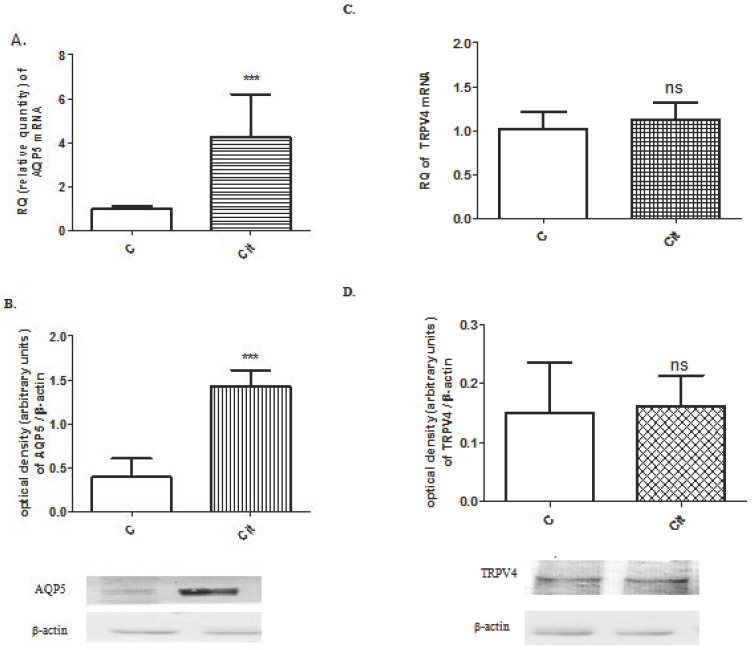
Changes in AQP5 mRNA (**A**) and protein (**B**, 35 kDa) and TRPV4 mRNA (**C**) and protein (**D**, 105 kDa) expression of on gestational day 22 in pregnant rat uteri after citral treatment, in vitro. β-actin (43 kDa) was used as a loading control. *C*: control uterus, *Cit*: citral-treated uterus, ns > 0.05; *** *p* < 0.001; compared to the control uterus sample. *n* = 6.

**Figure 4 life-11-00897-f004:**
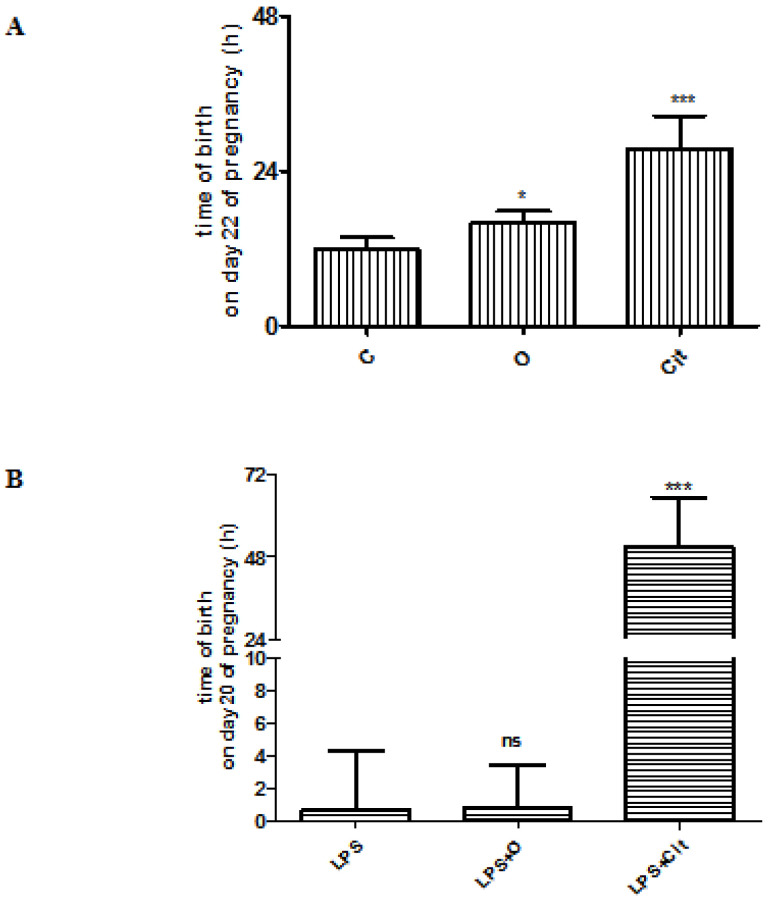
(**A**) Changes in the time of birth of normal pregnancy in *C* control, O: olive oil-treated and *Cit*: citral-treated rats. (**B**) Changes in the time of birth in bacterial lipopolysaccharide (*LPS*)-induced preterm birth, bacterial lipopolysaccharide and olive oil (*LPS + O*) or bacterial lipopolysaccharide and citral (*LPS + Cit*)-treated animals. ns > 0.05, * *p* < 0.05, *** *p* < 0.001; compared to the control or LPS-treated animals.

**Table 1 life-11-00897-t001:** Treatment protocol of late-pregnant rats. LPS: bacterial lipopolysaccharide; P.O.: orally; I.P.: intraperitoneally.

Group	Treatment	Daily Dose	Dosing Route
**1**	-	-	-
**2**	olive oil	1 mL	P.O.
**3**	citral	240 mg/kg	P.O.
**4**	LPS	50 μg/animal	I.P.
**5**	LPS + citral	50 μg/animal LPS + 240 mg/kg citral	I.P. + P.O.

## Data Availability

The data that support the findings of this study are available on request from the corresponding author.

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
