# Peer review of "The Effect of Citral on Aquaporin 5 and Trpv4 Expressions and Uterine Contraction in Rat—An Alternative Mechanism"

_life, 2021, doi:10.3390/life11090897_

Round 1
Reviewer 1 Report
This study provided interesting new insights into the potential roles of aquaporins and trpv channels in the regulation of preterm birth.
One major concern this reviewer has is that citral does not only have effects on TRPV4 and can influence signalling pathways, as stated in the discussion. Specific TRPV4 antagonists are available (e.g. GSK 2193874) which if utilised would have enhanced the significance of the findings presented in this study.
Please make sure this is more thoroughly addressed in the discussion.
Additional comments:
-how old were these animals?
-was this their first pregnancies?-in the in vivo studies - investigation of the length of the gestational period section of the methods please state that olive oil was used as a vehicle control.
-were no reverse transcription and no template controls utilised in the qPCR experiment?
-Please provide protein extraction protocol information.
-More information needed for western blot protocols: were the membranes blocked? How long were the membranes incubated in primary and secondary antibodies and at what temperature? Please provide secondary antibody information. Were technical replicates performed of the western blots or were they just performed once?
-Please merge figures 2 and 3. Include n values in figure legends.
-Please provide band size obtained in western blot figure legends.
-Immunohistochemical staining following incubation periods to assess whether the localisation of these molecules changed would have been a valuable addition to the manuscript.
-Similarly, quantification of calcium in the protein samples used for western blots would maybe provide interesting insights.
-As many TRPV channels are present in the uterus, alongside PMCAs etc, investigation into how citral influences the expression of other related molecules would have been a valuable addition to the manuscript, especially as the authors note that citral has a 'short-term blockage of TRPV4' whereas it can induce sustained inhibition of other TRP channels.
Author Response
"Please see the attachment."

Reviewer 2 Report
This study aims to determine the role of transient potential vanilloid 4 (TRPV4) in the regulation of pregnant uterine contraction. Particularly, in vitro uterine contractions were evoked by KCl, modifying the response with citral. The expressions of TRPV4 and Aquaporins 5 (AQP5) were measured by RT-PCR and Western blot techniques. Based on their data they determined that the AQP5 expression was significantly increased after citral incubation, while TRPV4 expression did not show significant changes.
In the material and methods section, I suggest inserting a figure summarizing the experimental model. It could be useful for the readers.
The discussion section should be improved. The authors cited and compared the results of the present study with their previous results, missing to compare their data with the international literature. Please, improve this section. Finally, I suggest improving the conclusion, inserting new ideas for future studies.
Author Response
"Please see the attachment."

Round 2
Reviewer 1 Report
Thank you for your detailed responses.